# Sex-Related Differences in Murine Models of Chemically Induced Pulmonary Fibrosis

**DOI:** 10.3390/ijms22115909

**Published:** 2021-05-31

**Authors:** Pavel Solopov, Ruben Manuel Luciano Colunga Biancatelli, Christiana Dimitropoulou, John D. Catravas

**Affiliations:** 1Frank Reidy Research Center for Bioelectrics Old Dominion University, Norfolk, VA 23508, USA; psolopov@odu.edu (P.S.); rcolunga@odu.edu (R.M.L.C.B.); cdimitro@odu.edu (C.D.); 2School of Medical Diagnostic & Translational Sciences, College of Health Sciences, Old Dominion University, Norfolk, VA 23508, USA

**Keywords:** idiopathic pulmonary fibrosis (IPF), nitrogen mustard, hydrochloric acid, gender differences, heat shock proteins, fibrosis

## Abstract

We developed two models of chemically induced chronic lung injury and pulmonary fibrosis in mice (intratracheally administered hydrochloric acid (HCl) and intratracheally administered nitrogen mustard (NM)) and investigated male–female differences. Female mice exhibited higher 30-day survival and less weight loss than male mice. Thirty days after the instillation of either HCl or NM, bronchoalveolar lavage fluid displayed a persistent, mild inflammatory response, but with higher white blood cell numbers and total protein content in males vs. females. Furthermore, females exhibited less collagen deposition, milder pulmonary fibrosis, and lower Ashcroft scores. After instillation of either HCl or NM, all animals displayed increased values of phosphorylated (activated) Heat Shock Protein 90, which plays a crucial role in the alveolar wound-healing processes; however, females presented lower activation of both transforming growth factor-β (TGF-β) signaling pathways: ERK and SMAD. We propose that female mice are protected from chronic complications of a single exposure to either HCl or NM through a lesser activation of TGF-β and downstream signaling. The understanding of the molecular mechanisms that confer a protective effect in females could help develop new, gender-specific therapeutics for IPF.

## 1. Introduction

Idiopathic pulmonary fibrosis (IPF) is a devastating disease, characterized by the progressive substitution of the lung parenchyma with fibrotic tissue and associated with poor prognosis, as reflected in the estimated mean survival of 2–5 years from the time of diagnosis [1]. Pirfenidone and Nintedanib, the only antifibrotic FDA-approved drugs that are recommended by the ATS/ERS/JRS/ALAT guidelines to treat IPF, can only reduce disease progression [2], and as of today a cure is still missing.

The precise pathophysiology of IPF is largely unknown; however, prevailing hypotheses assume a deficient wound-healing response to alveolar cell injury [3], genetic predisposition, disproportional immune response, and increased fibroblast activation [4]. Other theories consider gastroesophageal reflux and micro-aspiration of acid droplets a potential pathophysiological mechanism of IPF [5]. Indeed, acid-reducing medications have shown beneficial effects in some observational studies [6].

IPF has been associated with diverse blood and cellular biomarkers, such as chemokines (IL-8, CCL18), proteases (matrix metalloproteinases (MMP-1and MMP-7)), and growth factors (IGBPs) [7]. Indeed, bronchoalveolar lavage fluid (BALF) of humans diagnosed with IPF, shows increased values of MMP-3, -7, -8, and -9, which correlate with increased permeability index, rapid declining disease at 1 year and death [8]. Besides a few animal species, such as dogs, horses, and cats, which suffer from similar spontaneous pulmonary fibrotic disease, the development of proper animal models for the understanding of IPF pathophysiology, remains challenging [8]. The American Thoracic Society [9] recommends the use of intratracheally-instilled Bleomycin (BLM) to create the corresponding murine model of IPF.

Bleomycin is an antibiotic glycopeptide used for the treatment of solid cancers and lymphoma, which exerts its cytotoxic effects through both single and double-stranded DNA damage. After instillation, BLM provokes alveolar and cellular necrosis, interstitial edema, persistent inflammation, macrophage migration, hyperexpression of profibrotic markers, such as transforming growth factor-β (TGF-β) and the progressive deposition of a fibrotic scar [10]. Despite the current recommendation, the BLM-induced pulmonary fibrosis model has been criticized as not representative of IPF, mainly because of its rapidity of development and self-resolution after 21–28 days following BLM instillation [8].

Some countries consider IPF a rare disease, however in the last decades several studies reported increasing trends in mortality [11]. In the USA, 61.5 and 54.5 deaths per 100,000 men and women, respectively, are attributed to IPF every year [12]. The sex disparity in the mortality curves suggests a gender-related protective role of estrogens, however, investigations have produced inconclusive and controversial results.

To investigate gender-related differences in chemically induced pulmonary fibrosis, we intratracheally instilled male and female mice with either 2 μL/g body weight of 0.1 N hydrochloric acid (HCl) or 0.625 mg/kg body weight mechlorethamine, a nitrogen mustard alkilant. As we have published, these treatments provoke the development of acute alveolar inflammation, chronic lung injury, and pulmonary fibrosis, which last for at least 30 days [13,14,15].

Similar to BLM, mechlorethamine damages DNA strands by alkylating guanine in position 7, and promoting inflammation and cell apoptosis [16]. NM causes injury of both the upper and the lower airways with increased inflammation and increased immunostaining of COX-2, iNOS, and MMP9 [17]. NM instillation in rats elicits a pro-fibrotic response mediated by the migration of type 2 macrophages [18].

Acutely, HCl instillation provokes a strong inflammatory response mediated by the hyperexpression of tumor necrosis factor α (TNFα), MCP-1, IL-1β, and increased number of macrophages in the bronchoalveolar lavage fluid (BALF) [19]. The low pH of HCl (~1–2) corrodes alveolar structures, causing acute respiratory distress syndrome (ARDS), and persistent inflammation, with evidence of fibrous deposition appearing as early as 15 days after instillation [20]. Its use as a model of chemically-induced pulmonary fibrosis is supported by the hypothesis that acid micro-aspiration could play a role in the pathophysiology of IPF [5]. Additionally, human exposure to HCl or NM has been related to both acute intoxication and chronic lung disease [17]. Importantly, NM induces pulmonary fibrosis in mice without evidence of bronchiolitis obliterans syndrome, which is observed with the instillation of sulfur mustards, but it is not typically observed as a clinical characteristic of IPF [14,21].

Here we contrast the pathological profiles of mice 30 days after either HCl or NM instillation, to characterize possible sex-related pathways involved in the fibrotic process of the lung.

## 2. Results

### 2.1. Sex Differences in HCl- and NM-Induced Weight Loss and Mortality

Male and female mice instilled with saline, HCl, or NM displayed significant differences in outcomes. Male mice treated with NM showed a 30-day mortality rate of 20% while females exhibited only 15% mortality. Animals exposed to HCl displayed a mortality rate of 9% and 2% for male and female mice, respectively. Control male and female mice instilled with saline exhibit 100% survival (Figure 1A).

Changes in body weight were observed as early as 24 h after instillation of HCl, NM, or saline in both males and females. HCl-instilled males showed a significant and prolonged decrease in body weight when compared to saline-treated animals (*p* < 0.0001), while, following an initial decline in body weight, females recovered completely by day 5 (Figure 1B). NM-treated animals displayed a similar reduction in body weight that lasted throughout the observational period (Figure 1C). However, from day 23 on, females started to gain weight and recovered compared to males (*p* < 0.05).

### 2.2. Pulmonary Fibrosis Is Attenuated in Female Compared to Male Mice after Exposure to NM and HCl

Thirty days after the instillation of HCl or NM, severe lung histological abnormalities were observed when compared to saline-treated mice. Males instilled with HCl showed collagen deposition and large fibrous masses that were more pronounced than the corresponding females, while NM-instilled male animals displayed the highest evidence of fibrosis, as indicated by Masson’s trichrome staining of lung sections, with loss of parenchymal architecture and aberrant collagen deposition (Figure 2A). The degree of pulmonary fibrosis was also evaluated with the Ashcroft score [22], which was significantly higher in the male groups of either NM- or HCl-treated mice (Figure 2B). Ashcroft scores were confirmed by changes in the protein expression of collagen type I analyzed by western blotting of lung homogenates at 30 days post NM or HCl instillation. HCl- and NM-exposed males, displayed increased collagen levels when compared to HCl- and NM-instilled females (*p* < 0.01, *p* < 0.05) (Figure 2C,D).

### 2.3. Sex-Dependent Protection from Chemical Exposure Is Mediated by Lower Alveolar Inflammation

The instillation of either HCl or NM elicited a persistent inflammatory response that lasted till day 30 as reflected in increased concentrations of cells and proteins in the bronchoalveolar lavage fluid (BALF). For both chemicals, male animals displayed the highest increase in lung cellular infiltration and proteinosis when compared to female animals (Figure 3A,B, *p* < 0.001). Indeed, female mice showed just a modest increase in proteins and cellularity when compared to male animals for both chemicals (Figure 3A,B). Male mice, instilled with either HCl or NM, demonstrated a dramatic elevation in IL-1α levels, while female mice did not show any significant changes (Figure 3C). IL-1β levels increased in both sexes following NM (*p* < 0.001) again with males exhibiting much higher increases than females (Figure 3D). Transforming Growth Factor-β1 (TGF-β1), a crucial cytokine in the fibrotic process, was elevated at 30 days in HCl-treated males only (Figure 3E).

### 2.4. TGF-β Intracellular Profibrotic Pathways Are Mitigated in Female Mice

TGF-β mediates epithelial to mesenchymal transformation and the production of fibrotic matrix, through SMAD and non-SMAD signaling pathways. Thus, TGF-β1, ERK, and SMADS were analyzed in female and male mice 30 days after instillation with either NM or HCl. Male mice, instilled with HCl, demonstrated higher expression levels of TGF-β1 than corresponding saline controls, while female animals did not show any significant increase (Figure 4A). Female mice instilled with either saline or HCl exhibited significantly (*p* < 0.001) lower levels of activated ERK, one of the kinases in the non-SMAD canonical pathway of TGF-β (Figure 4B). The canonical signaling pathways of TGF-β, mediated by the progressive phosphorylation of Smad proteins and their nuclear translocation, showed increased activated levels of Smad3 in males (*p* < 0.05), but not in females (Figure 4D). Similarly, sex-related differences were found with the activation (phosphorylation) of Heat Shock Protein 90 (HSP90), a crucial chaperone involved in protein folding (*p* < 0.05; Figure 4E). In addition, matrix metalloproteinase 8 (MMP8) levels increased in the HCl male group, compared to female HCl-instilled or saline-instilled groups (*p* < 0.05; Figure 4F).

Male mice instilled with NM did not show activation of TGF-β1 but demonstrated a more pronounced involvement of its intracellular signaling, compared to females (Figure 5A). ERK activation (p-ERK/ERK levels) significantly increased 30 days after NM exposure, relative to either females or saline controls (*p* < 0.001; Figure 5B). Activation of SMAD-2 and SMAD-3, cascade kinases of the canonical pathway of TGF-β signaling, was prominent in male mice, while female animals showed almost no increase (*p* < 0.01; Figure 5C,D). HSP90 activation (p-HSP90/HSP90) also increased in male mice when compared to controls (*p* < 0.001) and in NM-instilled females only when compared to female controls (*p* < 0.05; Figure 5E). MMP8 levels did not change significantly in any group (*p* > 0.05; Figure 5F).

### 2.5. Females Are Protected against the Development of HCl and NM-Induced Lung Dysfunction

Changes in lung mechanics were analyzed 30 days after HCl- or NM-instillation. Male animals instilled with HCl showed a downward shift in pressure-volume (PV) loops reflecting stiffer lungs while females did not show significant changes from baseline (*p* < 0.05; Figure 6A). Both male and female NM-instilled mice displayed changes in PV loops compared to saline controls (*p* < 0.001, *p* < 0.05; Figure 6B). Total respiratory system resistance (Rrs), elastance (Ers), tissue damping (G), and tissue elastance (H) increased in HCl- and NM-instilled male mice, but not in the corresponding females (*p* < 0.05, *p* < 0.01, *p* < 0.001; Figure 6C,D,G,H). In addition, static compliance (Cst) and respiratory system compliance (Crs) were reduced in HCl- and NM-instilled male mice (*p* < 0.05, *p* < 0.001). It’s noteworthy that gender-related differences were present also in certain basal parameters, such as a reduced static compliance (*p* < 0.01) (Figure 6F) and a downward shift in PV loops (Figure 6A,B).

## 3. Discussion

We investigated possible sex-related pathways in two murine models of pulmonary injury and fibrosis. Instillation was preferred over inhalation for a better quantification of the dose of the administered chemical and thus better reproducibility of results. We have previously discussed the human equivalent dose computed from these models of HCl and NM-instillation. In addition, the chemicals were introduced into the caudal part of the trachea and pushed into lower bronchial structures by flushing air into the lungs. This allowed for a strong model of chronic lung injury and pulmonary fibrosis with a relatively high dose of NM but low mortality. The instillation of either HCl or NM was associated with a stronger weight loss and higher mortality in male than in female mice (Figure 1). Additionally, females exhibited a lower degree of fibrosis, as evidenced by architectural changes, Ashcroft score values and collagen expression (Figure 2). These data agree with observations in humans, where IPF shows less disease progression in women than in men [23].

As we have reported, instillation of either HCl or NM, in addition to acute robust inflammation, elicits a late milder inflammatory response in the lungs that is evident for at least 30 days [14,15]. In the present study, WBC and proteins were also elevated in BALF, but with male animals exhibiting stronger and more persistent inflammation (Figure 3). This was reflected in the BALF levels of IL-1α, IL-1β and TGF-β1, as well, which were much lower in females than males (Figure 3). Isoform 1 of TGF-β was also analyzed in lung homogenates by real-time qPCR. Male mice instilled with HCl, demonstrated a dramatic increase in TGF-β1 expression levels compared to their corresponding saline controls, while female animals did not show significant changes (Figure 4). Differently, no changes in the expression of TGF-β1 were observed in NM-instilled animals (Figure 5).

We then investigated crucial intracellular pathways involved in TGF-β fibrotic signaling in the lung. TGF-β signaling includes both canonical and non-canonical SMAD- and non-SMAD-pathways involving different kinases [24]. MAPK/ERK is a crucial kinase whose activity regulates the intracellular signaling of extracellular proteins and hormones and which, through the phosphorylation of transcriptional factors as C-myc, CREB, or C-*fos* promotes the synthesis of fibrotic proteins in response of external stimuli [25]. MAPK/ERK phosphorylation has been implicated in the non-SMAD TGF-β signaling playing a key role in the profibrotic response [26]. HCl-treated male mice showed a significant activation of the non-SMAD and SMAD pathways, as reflected in ERK and SMAD3 phosphorylation (Figure 4). Similarly, NM-treated male mice showed higher activation of SMAD2, SMAD3, and ERK compared to females (Figure 5).

The expression of activated HSP90 (phosphorylated HSP90) increased in both male and female mice instilled with NM compared to their corresponding saline controls (Figure 5). However, only male mice instilled with HCl exhibited increased levels of P-HSP90 (Figure 4). Heat shock proteins (HSPs) act as co-chaperones, responsible for the correct synthesis, folding, and assembling of “client proteins” and stabilizing several intracellular pathways. Overexpression of HSP90 in the lung is associated with the wound-healing response [27]. HSP90 is overexpressed in fibroblasts of patients with pulmonary fibrosis [28], is considered a pro-fibrotic marker and a possible target of new therapies to counteract IPF [29,30,31].

Matrix metalloproteinases (MMPs) are aberrantly expressed during the development of IPF. They destroy components of the extracellular matrix and cleave and activate a wide range of growth factors, cytokines (especially IL-10), chemokines and cell surface receptors [30]. They affect numerous cell functions including adhesion, proliferation, differentiation, recruiting, transmigration and apoptosis in the lung [31]. MMP-8 expression in lung homogenates was higher in HCl and NM-instilled male mice when compared to saline and females (Figure 4 and Figure 5). Interestingly, a similar sex-related difference was observed in humans with chronic lung injury due to tuberculosis infection, with males exhibiting higher blood values of MMP8 than females [32].

The changes observed by molecular and histological evaluation were confirmed by analysis of lung mechanics (FlexiVent), where the level of lung dysfunction was assessed at 30 days post-instillation. While males showed a downward shift in pressure–volume (PV) loops, females did not show significant changes. In addition, males exhibited increases in total respiratory system resistance (Rrs), elastance (Ers), tissue damping (G), and tissue elastance (H) for both HCl and NM, which were not present in the corresponding female animals (Figure 6). Results from a bleomycin-induced lung injury mouse model show that lung function declines independently of estrogen in both male and female mice [33]. Our findings are consistent with a study in which lung function parameters in patients with PF, such as forced vital capacity, (FVC% of predicted) and total lung capacity (TLC% of predicted), were significantly lower in men compared to women [34]. Riches et colleagues [35] reported that female mice, after 14 days of BLM-instillation, showed lower inflammation, collagen, and hydroxyproline deposition, and an overall lower degree of pulmonary fibrosis that they attributed to a less tonic signaling pathway of TGF-β by the misplaced nuclear localization of phospho-SMAD2/3. A silica-induced model of pulmonary fibrosis displayed a stronger inflammation in female mice at day 14, with higher cellularity and cytokine activation, but a lower degree of hydroxyproline, collagen deposition, and fibrosis. Ovariectomized female mice exhibited reduced inflammatory markers in the lung (total cells, macrophages, lymphocytes, MCP-1, CCL9) compared to control females, but restored female-sensitivity to silica with a higher hydroxyproline total content [36]. BLM-instilled female mice showed a lower decline in static compliance when compared to male mice, an effect that was reversed by castration in males or by virilization in female animals [33]. Taken together these results indicate that estrogens and androgens influence the pattern of both lung inflammation and fibrosis.

Our data indicate that female mice are protected from the development of chronic lung injury mediated by either HCl or NM and suggest that this protection could stem from differences in TGF-β1 signaling. Indeed, NM-instilled animals did not show significant increases in TGF-β1 expression analyzed either in the BALF or in tissue. However, as both HCl- and NM-induced pulmonary fibrosis models exhibited a sex-related activation of canonical SMAD2 and SMAD3, suggesting that other isoforms of TGF-β are activated when mice are exposed to NM. For example, TGF-β3 mediates tissue repair, but also is able to downregulate TGF-β1-induced gene expression; at the same time, TGF-β3 overexpression results in an upregulation of Smad proteins similar to TGF-β1 [37]. On the other hand, the non-SMAD pathway of TGF-β activates RAS, Raf, MEK, and ERK, which exert its action on different transcriptional factors through CREB phosphorylation [38]. It is known that ERK exhibits sex related activation. Female rats display lower ERK activation compared to males and this protects animals from interleukin 1β-induced abdominal aortic aneurisms by preventing the activation and synthesis of MMPs [39]. This mechanism is supported by Wu and colleagues [40], which showed how estrogen therapy protects against aortic aneurism formation through diminishing MMP2 and MMP9. ERK plays a crucial role in inflammation by promoting leukotriene synthesis, a mechanism that is regulated by androgens and inhibited by estrogens [41]. When ERK is activated, it phosphorylates CREB and promotes synthesis of several proteins, such as MMPs. Mice lacking MMP8 are protected from bleomycin-induced pulmonary fibrosis and their fibroblasts show higher IL-10 levels and lower collagen synthesis, compared to those of wild type mice [42]. BALF of humans diagnosed with IPF exhibits increased levels of MMP-3, -7, -8 and -9, which correlate with increased permeability index, rapid declining disease at 1 year and death [43]. Similarly, patients diagnosed with IPF exhibit sex disparity in MMP-7 gene expression when compared to controls or to patients suffering from chronic obstructive pulmonary disease (COPD) [44]. This response is mediated by the overexpression of activated HSP90, which stabilizes these proteins by promoting their correct folding, as shown in Figure 7.

Our manuscript underlined various histological, molecular, and mechanical differences between males and females exposed to pro-fibrotic chemicals. The association of new sex-related molecular mechanisms involved in the fibrotic process of the lung could improve our understanding of potential new personalized therapeutic approaches for IPF and suggest the usefulness of gender-based subgroup analyses of published large interventional studies.

## 4. Materials and Methods

### 4.1. Materials

Hydrochloric acid (ACS grade), methacholine (USP grade), red protein G affinity gel beads, RIPA buffer, and protease inhibitor cocktail were obtained from Sigma-Aldrich Corporation (St. Louis, MO, USA). Socumb (pentobarbital) USP grade, Anased (xylazine) USP grade and Ketaset (ketamine) USP grade were supplied by Henry Schein Animal Health (Pittsburg, PA, USA). 10% formaldehyde, PureLink^TM^ DNase Set, RNase Inhibitor and RNAlater were purchased from Thermo Fisher Scientific (Waltham, MA, USA), the BCA Protein assay kit from Pierce Co. (Rockford, IL, USA), EDTA and Western blot membranes from GE Healthcare (Chicago, IL, USA), TRIzol^®^ and SuperScript^TM^ IV VILO Reverse transcription Kit from Invitrogen (Carlsbad, CA, USA), RNeasy mini kit from Qiagen (Hilden, Germany) and SYBR Green Master Mix from Applied Biosystems (Carlsbad, CA, USA). TGF-b1 and β-actin primers (TGF-b1 F: 5′-GGA CGA GCT GGT TGA GAG AA-3′; TGF-b1 R: 5′-GCA GTA GCA GCG GAA AAG TC -3′; Beta actin F: 5′-CCC CTG AGG AGC ACC GTG TG -3′; Beta-actin R: 5′-ATG GCT GGG GTG TTG AAG GT-3′) used for real time quantitative PCR were purchased from Integrated DNA technologies, Inc. (Coralville, IA, USA). All antibodies used in Western blots have published immunospecificity data available online. For antibodies used in Western blots, rabbit total and phosphorylated ERK44/42, SMAD2, SMAD3, MMP8, and HSP90 were obtained from Cell Signaling Technology, Inc. (Danvers, MA, USA), mouse monoclonal anti-β-Actin from Sigma-Aldrich Corporation, Collagen type I rabbit antibody from Thermo Fisher Scientific (Waltham, MA) and IRDye 800CW Goat anti-rabbit and IRDye 680RD Goat anti-mouse, NewBlot PVDF Stripping Buffer from LI-COR Biosciences (Lincoln, NE, USA). For preparation of SDS-PAGE: ProtoGel (30% acrylamide mix) and TEMED were from National Diagnostics (Atlanta, GA, USA), Tris-HCl buffer from Teknova (Hollister, CA, USA), 10% SDS, and ammonium persulfate from Thermo Fisher Scientific, Protein dual color standards, and Tricine Sample Buffer were purchased from Bio-Rad Laboratories (Hercules, CA, USA).

### 4.2. Animals and Treatment Groups

All animal studies were approved by the Old Dominion University IACUC and adhere to the principles of animal experimentation as published by the American Physiological Society. Male and female pathogen-free C57Bl/6J (Jackson Laboratories, Bar Harbor, ME, USA) mice (8–10 weeks old, 20–28 g weight) were randomly divided into six treatment groups: (1) Male control group: received 2 μL/g body weight saline intratracheally (i.t.); (2) female control group: received 2 μL/g body weight saline (i.t.); (3) and (4) male and female mice that were exposed to 2 μL/g body weight of 0.1N HCl (i.t.); (5) and (6) male and female mice that were exposed to 0.625 mg/kg body weight mechlorethamine hydrochloride (i.t.) (*n*  =  12 mice per group). All analyses were performed at 30 days post chemical exposure. In order to perform intratracheal administration of hydrochloric acid, mechlorethamine or saline, mice were first anesthetized with intraperitoneal injections of xylazine (6 mg/kg) and ketamine (60 mg/kg). Secondly, an i.p. bolus of sterile normal saline (10 μL/g) was given as pre-emptive fluid resuscitation. A small neck skin incision (~1 cm) and separation of the salivary glands was made to visualize the trachea, and while mice were suspended vertically, a fine 20 G for male or 22 G for female plastic catheters were introduced through the mouth and the cannulation of the trachea was confirmed by visualization of the catheter from the open neck incision. Then, freshly prepared hydrochloric acid solution (groups 3,4) or mechlorethamine hydrochloride (groups 5,6) or sterile saline 0.9 % NaCl solution (groups 1,2) was introduced (2 μL/g body weight) and flushed with 100 μL air. Finally, the catheter was withdrawn, the neck incision was sutured by surgical adhesive and animals were placed in ventral position on top of a heating pad, under supplemental oxygen (slowly weaned from 100 to 21% O_2_), and observed for the next four hours for signs of respiratory distress. Mice were later returned to their home cages and monitored daily for abnormal physical appearances. After 30 days, all groups were euthanized, and bronchoalveolar lavage fluid and whole lung tissue were collected for analysis.

### 4.3. Lung Mechanics Measurements

Mice from all groups were anesthetized with pentobarbital (90 mg/kg, i.p.), tracheostomized with a metal 1.2 mm (internal diameter) cannula, and connected to a FlexiVent small animal ventilator (SCIREQ Inc., Montreal, QC, Canada). Ventilation was performed at a tidal volume of 10 mL/kg and respiratory rate of 150/min. A 15-min stabilization period was allowed before any measurements began. Firstly, following a deep inflation, resting static compliance (Cst, mean of 3 values) and pressure-volume relationships (PV curves) were estimated by stepwise increasing airway pressure to 30 cm H_2_O and then reversing the process. Both parameters reflect the intrinsic elasticity of the lungs and are either reduced (Cst) or shifted to the right (PV curves) in fibrosis. Secondly, Snapshot-150 and Quick Prime-3 maneuvers were performed. Total resistance (Rrs) and elastance (Ers), reflecting the behavior of the entire respiratory system (peripheral and conducting airways, chest wall and parenchyma), and Newtonian resistance (Rn) and tissue damping (G) values, the former reflecting resistance of the large, conducting airways, and the latter reflecting mostly parenchymal and peripheral airway contributions were calculated and are presented as a mean of 12 recordings.

### 4.4. Histopathology and Lung Injury Scoring

Immediately after euthanasia, mice were positioned in upright position and the lungs were instilled and inflated with 10% formaldehyde solution to a pressure of 15 cm H_2_O and then immersed in the same solution for at least 72 h. After fixing, lung tissue samples were embedded in paraffin. For collagen staining, sections about 5 μM thick were prepared from the paraffin blocks and stained with Masson’s trichrome. Twenty randomly selected fields from each slide were examined under 20× magnifications. All slides were scored according to the Ashcroft score method in order to estimate the severity of pulmonary fibrosis [22]. An investigator blinded to the study protocol performed the scoring.

### 4.5. Bronchoalveolar Lavage Fluid (BALF) White Blood Cells Count

Bronchoalveolar lavage fluid (BALF) was collected by washing the lung with sterile 1X PBS (1 mL) via the tracheal cannula. The fluid was centrifuged at 2500× *g* for 10 min at 4 °C (Thermo Fisher Centrifuge 5417R) and the supernatant was collected and stored at −80 °C. The cell pellet was resuspended in 1 mL sterile PBS and the total number of white blood cells was determined using a hemocytometer.

### 4.6. Total Protein and Cytokines Analysis in BALF

BALF supernatant was prepared as described in the previous section. Total protein concentration was determined using the micro bicinchoninic acid (BCA) assay according to the manufacturer’s protocol. BALF supernatant cytokines (interleukin (IL)-1a, IL-1b and human TGF-β) were analyzed in triplicate by the chemiluminescent Q-Plex ^TM^ technology (Quansys Biosciences, Logan, UT, USA).

### 4.7. Lung Tissue Collection

Immediately after euthanasia, the thorax was opened, blood was drained from the heart through the right ventricle and the pulmonary circulation was flushed out with sterile PBS containing EDTA. The lungs were dissected from the thorax, snap-frozen in liquid nitrogen and kept at −80 °C for subsequent analysis.

### 4.8. Western Blot Analysis

Proteins in lung tissue homogenates were extracted from frozen lungs by sonication (50% amplitude, 3 times for 10 s) in ice-cold RIPA buffer with added protease inhibitor cocktail (100:1). The protein lysates were gently mixed under agitation for 3 h at 4 °C, and then centrifuged twice at 14,000× *g* for 10 min. The supernatants were gathered, and total protein concentration was determined using the micro BCA assay. Equal amounts of proteins from all lysates were used for Western blot analysis. The samples were first mixed with Tricine Sample Buffer 1:1, boiled for 5 min and then separated on a 10–12% polyacrylamide SDS gel by electrophoresis. Separated proteins were then transferred to a nitrocellulose membrane, incubated with the appropriate primary antibody, followed by incubation with the secondary antibody and detected by digital fluorescence imaging (LI-COR Odyssey CLx, Dallas, TX, USA). Beta-actin was used as loading control. ImageJ software v.1.8.0 was used to perform densitometric quantification of the bands from the Western blot membranes (http://imagej.nih.gov/ij/; accessed on 27 May 2021; National Institutes of Health, Bethesda, MD, USA). For ERK and p-ERK, both bands were quantified together. Membranes were stripped in stripping buffer 20 min, blocked, and incubated with other primary and secondary antibodies.

### 4.9. RNA Isolation and Quantitative Real-Time PCR (qPCR)

Lung tissue, stored in RNAlater solution, was dried and homogenized in TRIzol^®^ followed by a cleaning up step using the RNeasy Mini Kit. The purified RNA was transcribed into cDNA using the SuperScriptTM IV VILO Reverse transcription Kit and analyzed by real-time qPCR with SYBR Green Master Mix on a StepOne Real-Time PCR System (Applied Biosystems v.2.3). Results were evaluated using the standard curve method and expressed as fold of control values. Beta-actin expression was used for the normalization of each mRNA expression levels for all samples. Specifically designed primer pairs and qPCR conditions were applied to selectively determine the expression of mouse beta-actin and TGF-b1 as previously described (Veres-Szekely et al. 2017).

### 4.10. Statistical Analysis

Statistical significance of differences among groups was determined by one way- or two way-analysis of variance (ANOVA) followed by the Tukey post-hoc test using GraphPad Prism Software (GraphPad Software, San Diego, CA, USA). Differences among groups were considered significant at *p* < 0.05.

## 5. Conclusions

Hydrochloric acid and nitrogen mustard are known toxic chemicals whose inhalations are related to dangerous chronic complications, such as pulmonary fibrosis. Sex-related differences were investigated and revealed lower alveolar inflammation, TGF-β synthesis, ERK, and SMAD phosphorylation, and lower production of MMP8 in females. Thirty days after instillation, female animals were protected from the deposition of collagen, development of lung dysfunction, and pulmonary fibrosis. The study of sex-related differences in the fibrotic process and the activation of different crucial kinases involved in TGF-β signaling pathways could improve our understanding of pulmonary fibrosis and unveil new personalized therapeutic approaches for idiopathic pulmonary fibrosis.

## Figures and Tables

**Figure 1 ijms-22-05909-f001:**
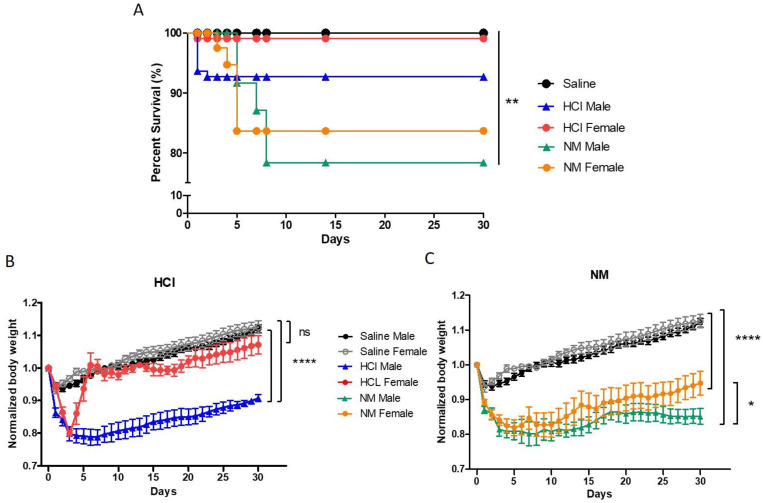
(**A**) Kaplan–Meyer survival curves after NM or HCl exposure in female and male mice (** *p* < 0.01) with log-rank (Mantel–Cox) test, *n* ≥ 25. (**B**) HCl provoked a significant decrease in body weight (20%) in both males and females at 48 h after instillation. Female animals recovered quickly and started gaining weight similarly to saline treated animals (ns: *p* > 0.05) while male mice maintained a reduced body weight different from either saline or females (**** *p* < 0.0001). One-way ANOVA with the Bonferroni post-test; *n* ≥ 6. (**C**) NM provoked a significant decrease in body weight (~20%) in both males and females at 48h after instillation, compared to corresponding saline-treated animals (**** *p* < 0.0001). From day 23 on, females began gaining weight at a faster rate than males (* *p* < 0.05). Normalized body weight data were obtained by dividing body weight of each animal by its day 0 (day of instillation) body weight. One-way ANOVA with the Bonferroni post-test; *n* ≥ 6.

**Figure 2 ijms-22-05909-f002:**
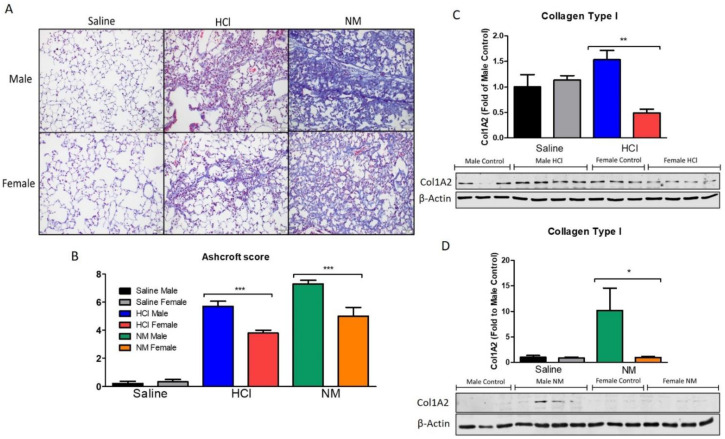
(**A**) Histological analysis of lung injury after intratracheal instillation of saline, NM, or HCl in male and female mice. Masson’s trichrome staining of lung sections at 30 days post-instillation. Male mice instilled with HCl displayed significant increase in perivascular collagen and loss of parenchymal architecture compare to control. Female mice showed formation of collagen fibers without severe breaking of parenchymal structures. Male mice receiving NM demonstrate increased parenchymal and perivascular collagen deposition and large areas with fibrous obliteration, while female NM-treated mice exhibited significantly less collagen staining. Scale bar: 100 µM. (**B**) Ashcroft scores of lung fibrosis; male mice have higher scores compared to females in both models of chemical exposure. (**C**,**D**) Collagen type I protein levels in female and male mice treated with either HCl (**C**) or NM (**D**) were analyzed by western Blotting. Male animals showed the highest collagen deposition in the lungs. Means ± SEM; *** *p* < 0.001; ** *p* < 0.01, * *p* < 0.05 with one-way ANOVA and Tuckey’s, *n* = 6.

**Figure 3 ijms-22-05909-f003:**
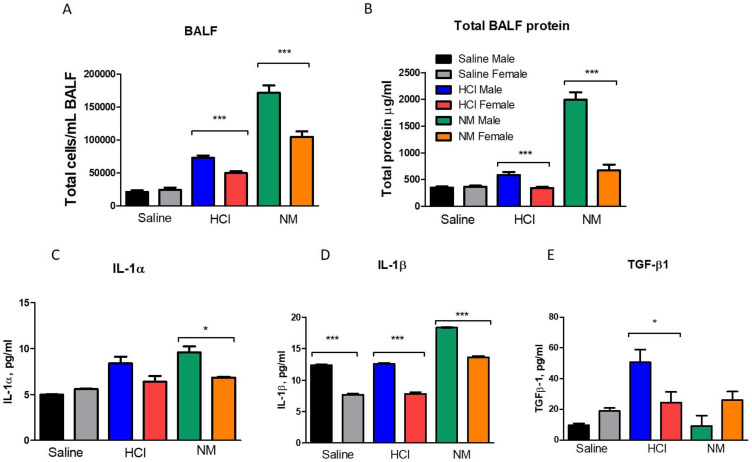
Analysis of Bronchoalveolar lavage fluid (BALF). (**A**) Total cell count; (**B**) total protein concentrations; (**C**) IL-1α; (**D**) IL-1β; and (**E**) TGF-β1. Male and female mice were instilled with HCl or NM. Means ± SEM, * *p* < 0.05; *** *p* < 0.001 with one-way ANOVA and Tuckey’s, *n* =6.

**Figure 4 ijms-22-05909-f004:**
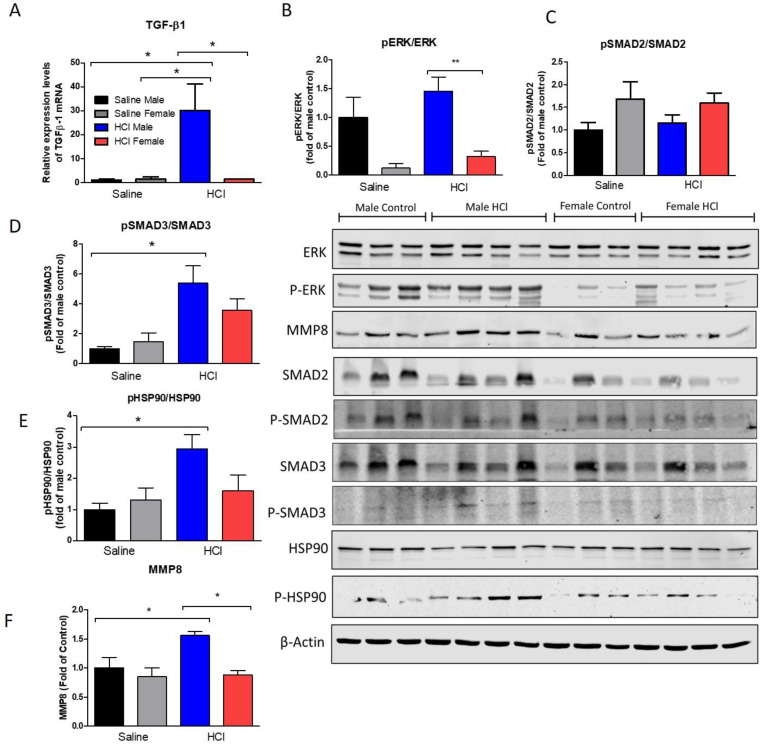
Intracellular pathways of TGF-β 30 days after HCl instillation. (**A**) Male mice treated with HCl displayed increased gene expression levels of TGF-β1 compared to saline or females. (**B**) Male animals showed high levels of p-ERK compared to females 30 days after HCl instillation. (**C**) The phosphorylated form of Smad3 was significantly increased in males but not in female animals. (**D**) Heat Shock Protein 90 activation (pHSP90) increased only in males. (**F**) Matrix metalloproteinase levels were higher in males (**E**). TGF-β1 mRNA expression levels were measured by real-time qPCR and normalized to β-actin. Band density was normalized to β-actin and plotted as fold of control. Means ± SEM; ** *p* < 0.01, *p* < 0.05 with one-way ANOVA and Tuckey’s, *n* = 3–4.

**Figure 5 ijms-22-05909-f005:**
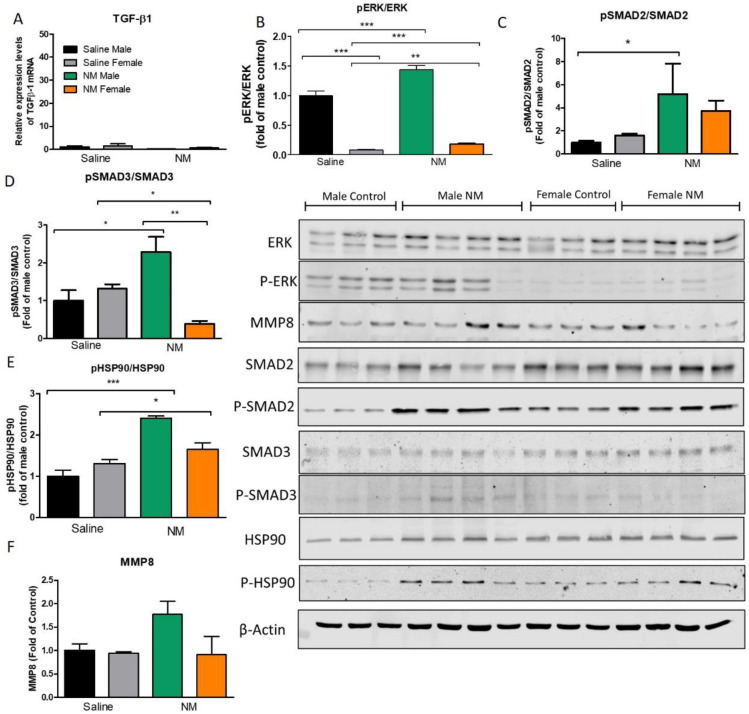
Intracellular pathways of TGF-β activation 30 days after NM instillation. (**A**) No changes were observed in the expression levels of isoform 1 of TGF-β between males and females instilled with NM. (**B**) Male mice treated with NM displayed increased levels of p-ERK compared to females. (**C**,**D**) SMAD2 and SMAD3 were activated only in males. (**E**) Heat Shock Protein 90 activation (pHSP90) increased only in males. (**F**) Matrix Metalloproteinase levels were not significantly altered. TGF-β1 mRNA expression levels were measured by real-time qPCR and normalized to β-actin. Band density was normalized to β-actin and plotted as fold of control. Means ± SEM; *** *p* < 0.001, ** *p* < 0.01, * *p* < 0.05 with one-way ANOVA and Tuckey’s, *n* = 3–4.

**Figure 6 ijms-22-05909-f006:**
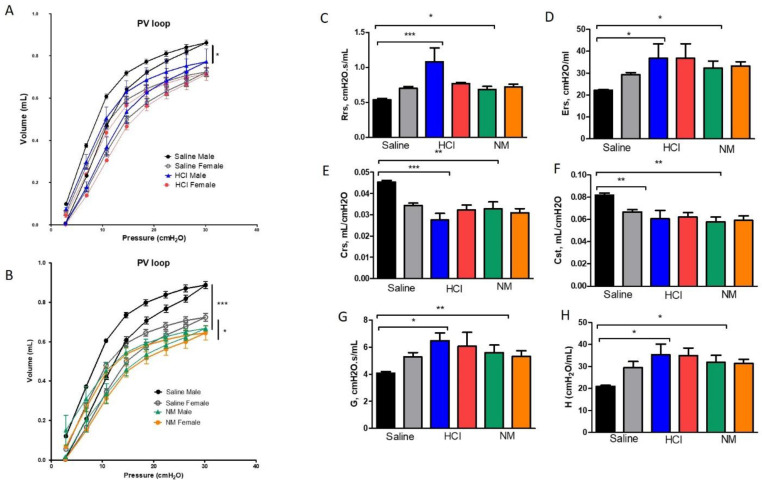
Gender differences in HCl and NM-induced alterations in lung mechanics. (**A**,**B**) Either HCl or NM provoked a downward shift in pressure-volume loops (PV loops), with the highest effect observed in male mice. (**C**,**D**,**G**,**H**) Male mice instilled with either HCl or NM, showed an increase in total respiratory system resistance (Rrs), total elastance (Ers), tissue elastance (**H**), and damping (**G**). (**E**,**F**) Respiratory system compliance (Crs) and static compliance (Cst) were reduced in HCl and NM-treated male mice. Means ± SEM; *n* = 3–4 mice per group; * *p* < 0.05, ** *p* < 0.01, *** *p* < 0.001, ns: not significant, with one-way ANOVA and Tukey’s.

**Figure 7 ijms-22-05909-f007:**
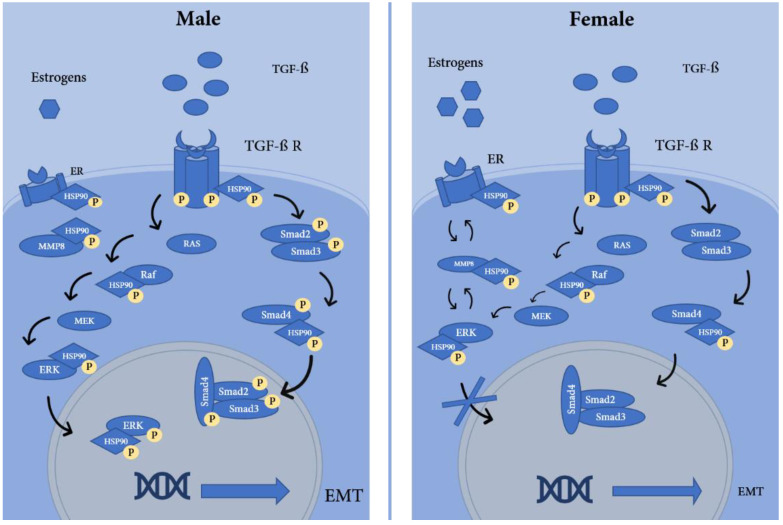
Schematic representation of the proposed gender related differences in TGF-β, SMAD- and non-SMAD-dependent pathways. The non-SMAD dependent pathway involves a large group of kinases, including RAS, MEK, and ERK. ERK is an indirect client protein of HSP90. The SMAD pathway involves different proteins of the same family which translocate to the nucleus. Both SMAD and non-SMAD dependent pathways promote the activation of pro-fibrotic transcription factors leading to the production of collagen, fibrosis and epithelial to mesenchymal transformation (EMT). In females, ERK shows limited activation and phosphorylation of the SMAD proteins, resulting in lessened nuclear transcription and attenuated fibrosis and EMT. Estrogen receptor has a lesser activity profile in men, and thus its interaction with ERK and MMP8 is proposed as a protective mechanism. The phosphorylated form of HSP90 mediates the proper stabilization of profibrotic proteins, thus represents an exciting target for IPF. Additional studies are needed to confirm our results. Immunofluorescence staining of lung sections will be useful in validating the phosphorylation status of Smad2/3 in mouse lungs. Experiments in ovariectomized female mice and castrated male mice would further clarify the role of sex in the development of pulmonary fibrosis, as would comparisons of the current findings with the bleomycin mouse model of PF.

## Data Availability

Not applicable.

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
