# Peer review of "Sex-Related Differences in Murine Models of Chemically Induced Pulmonary Fibrosis"

_ijms, 2021, doi:10.3390/ijms22115909_

Round 1

Reviewer 1 Report

The study describes the sex differences in HCl and NM induced pulmonary fibrosis
major: The canonical and non-canonical pathway has been presented in the results using whole lung tissue lysates. However, measuring the protein levels from the lung tissue homogenates does not give a comprehensive idea of the signalling pathway.

Major comments: The fibroblasts or epithelial cells should be used to elucidate the ERK and SMAD pathways.

Minor Comments:
- Line 115 in section 2.2 does not mention what assay was used to measure the collagen type I levels.
- Section 2.3 does not mention about WBC inflitration and proteosis in females during HCl or NM treatment

Author Response

Major: The canonical and non-canonical pathway has been presented in the results using whole lung tissue lysates. However, measuring the protein levels from the lung tissue homogenates does not give a comprehensive idea of the signalling pathway. The fibroblasts or epithelial cells should be used to elucidate the ERK and SMAD pathways. 

ERK and Smad proteins are crucial in TGF-B signalling. Fibroblasts, endothelial and epithelial cells have all been shown to respond to TGF-b in models of lung injury (https://www.nature.com/articles/s12276-019-0371-7; https://www.ahajournals.org/doi/10.1161/01.atv.0000256466.65450.ce; https://www.spandidos-publications.com/10.3892/mmr.2020.11453). Analysis of lung homogenates simultaneously evaluates endothelial, fibroblasts and epithelial/alveolar cell responses- as employed in this paper- is widely accepted and thus results are considered as valuable markers of disease progression. In this paper, we focused mainly on possible sex-related differences during chronic lung injury, without trying to define population-specific pathways. However, the reviewer points out a very interesting topic that we already plan to address in our future work.

Line 115 in section 2.2 does not mention what assay was used to measure the collagen type I levels.

We utilized Western blotting; this is now included.

Section 2.3 does not mention about WBC inflitration and proteosis in females during HCl or NM treatment

The manuscript has been corrected and now includes these observations.

Reviewer 2 Report

General Comments:

  1. The introduction should include additional rationale and clarification for the studies performed; phrasing is awkward. The dose of NM described is different in the introduction and the methods; this needs to be corrected.
  2. The NM dose used for the studies seems quite high; what is the biological relevance of this dose to human exposure? 
  3. Justification should be included for the intratracheal instillation of HCL and NM as this is an invasive procedure.
  4. Additional literature describing HCL and mustard vesicant-induced lung injury and fibrosis in animals would strengthen the manuscript, specifically as to how the current studies compare to published literature in terms of effects, doses  and routes of administration.
  5. The rationale for the use of HCl or NM as a model for IPF model is weak as etiology of IPF is unknown.
  6. Further clarification should be provided regarding lung function methodology. What is the rationale for a 15-minute rest period post snapshot-150? Multiple replicates should be used for the Snapshot-150/quick prime maneuver as responses can be variable. In addition, why were the  lung function studies performed separately?
  7. The discussion is fragmented. The authors focus on TGF- β signaling, but the only results provided are from the cytokine quantification in BALF. Furthermore, TGF- β was not significant in male or female mice following NM exposure. The manuscript would benefit from additional analysis of TGF- β.
  8. The SMAD and HSP90 western blot data are over interpreted. The discussion details SMAD-signaling as it related to both HCl and NM, but western blots for SMAD were only provided for NM exposed mice. There is no mention of the lung function data within the discussion.
  9. The conclusion should be revised as the majority of the data show that there are no changes in female mice.
  10. A more careful review of the paper is suggested, as there are some clerical errors.

Specific Comments:

Figure 1: the y-axis (“Normalized body weight”) of panels B and C is unclear. Authors should include how body weight was normalized.

Figure 2: No methods are included for Collagen Type 1 protein expression quantification.

Figure 3A: WBC is not an appropriate label for the figure, since differentials were not performed.  Statistics in figure 3 are overwhelming and give no clear message.

Figures 4 and 5:  HSP90, SMAD2, and SMAD3 western blots should be reproduced. Further clarification should be provided for the use of MMP-8 as biomarker; the introduction is specific to MMP-1 and MMP-7. Additional explanation is needed as to why there was no ERK expression in female mice. The densitometry method for analyzing the double bands in the ERK western blot should be explained. The legend in figure 5 does not define the statistics (***) presented for ERK and HSP90.

Figure 7: The data provided in the manuscript do not accurately support the graphic. Male mice also have estrogen receptors that were omitted in the figure.

All figure legends should include type of ANOVA performed (two-way vs. one-way).

Author Response

The NM dose used for the studies seems quite high; what is the biological relevance of this dose to human exposure? 

The dose employed is 0.625 mg/kg for mice which is similar the equivalent dose employed in rats (0.125 mg/kg) https://pubmed.ncbi.nlm.nih.gov/26243812/#:~:text=Nitrogen%20mustard%20(NM)%20is%20a,necrosis%20factor%20(TNF)%CE%B1. We have previously discussed the corresponding human equivalent dose of NM and provided evidence that this dose corresponds to the Temporary Emergency Exposure Levels (TEELs) established by the Department of Energy (DOE) that stands for “irreversible” and “life threatening” injury (https://pubmed.ncbi.nlm.nih.gov/32362214/.

Justification should be included for the intratracheal instillation of HCL and NM as this is an invasive procedure.

Instillation is a widely employed and recognized technique to study acute and chronic lung injury. This approach allows for a better defined, more accurate and better replicated dose. During surgery, after the intratracheal instillation, ~100 uL of air were flushed in order to push the chemical to the lower structures of the lungs. In addition, the instillation technique represents a safer choice of chemical delivery, compared to inhalation, for both researchers and environment. We have followed the reviewer’s comment and have modified the Discussion.

Additional literature describing HCL and mustard vesicant-induced lung injury and fibrosis in animals would strengthen the manuscript, specifically as to how the current studies compare to published literature in terms of effects, doses and routes of administration.

We have extensively discussed our model of HCL- and NM-instillation in previous papers (https://www.tandfonline.com/doi/abs/10.1080/08958378.2019.1624895?journalCode=iiht20; https://pubmed.ncbi.nlm.nih.gov/32362214/). However, additional information has been added to the introduction and discussion sections.

The rationale for the use of HCl or NM as a model for IPF model is weak as aetiology of IPF is unknown.

We thank the reviewer for this comment. Bleomycin is employed as a model of IPF even if the cause of IPF is still unknown. However, the bleomycin-induced fibrosis model has been severely criticezed because of its reversibility. We propose that HCl and NM may be at least equally valid models of pulmonary fibrosis, that further exhibit sex-related differences which are similarly observed in large clinical studies of IPF patients.

Further clarification should be provided regarding lung function methodology. What is the rationale for a 15-minute rest period post snapshot-150? Multiple replicates should be used for the Snapshot-150/quick prime maneuver as responses can be variable. In addition, why were the lung function studies performed separately?

We thank the reviewer for this question. There were unfortunate errors in the description these experiments in Methods. The section has been rewritten.

The discussion is fragmented. The authors focus on TGF- β signaling, but the only results provided are from the cytokine quantification in BALF. Furthermore, TGF- β was not significant in male or female mice following NM exposure. The manuscript would benefit from additional analysis of TGF- β.

We thank the reviewer for this question. While TGF-b1, the leading cytokine of the fibrotic process is overexpressed in HCl-instilled animals, it is not significantly changed at 30 days after NM instillation. This is in accordance with our previous published data, were TGF-B1 was increased modestly only at 10 days post-instillation of NM (https://pubmed.ncbi.nlm.nih.gov/32362214/). We have performed analysis of TGF-B1 gene expression levels in lung homogenates by real-time qPCR and have added them to the manuscript. As SMAD pathways are activated in NM-instilled animals, we hypothesize that other isoforms of TGF-B may be involved in that model. The Discussion section is also updated.

The SMAD and HSP90 western blot data are over interpreted. The discussion details SMAD-signalling as it related to both HCl and NM, but western blots for SMAD were only provided for NM exposed mice. There is no mention of the lung function data within the discussion.

Western blot data of the phosphorylation of Smad2 and Smad3 for HCl-instilled animals have been added to the manuscript. Flexivent data analysis has been added to the discussion.

The conclusion should be revised as the majority of the data show that there are no changes in female mice.

The conclusions have been modified.

A more careful review of the paper is suggested, as there are some clerical errors.

The manuscript has been carefully reviewed.

Specific Comments:

Figure 1: the y-axis (“Normalized body weight”) of panels B and C is unclear. Authors should include how body weight was normalized.

Body weight normalization was added to the legend.

Figure 2: No methods are included for Collagen Type 1 protein expression quantification.

Collagen was measured by Western blotting; bands have been added to the figure and methods were updated.

Figure 3A: WBC is not an appropriate label for the figure, since differentials were not performed.  Statistics in figure 3 are overwhelming and give no clear message.

Red blood cells in BALF were lysed, so the only cells counted are WBC. Statistic in figure 3 have been modified.

Figures 4 and 5:  HSP90, SMAD2, and SMAD3 western blots should be reproduced. Further clarification should be provided for the use of MMP-8 as biomarker; the introduction is specific to MMP-1 and MMP-7. Additional explanation is needed as to why there was no ERK expression in female mice. The densitometry method for analyzing the double bands in the ERK western blot should be explained. The legend in figure 5 does not define the statistics (***) presented for ERK and HSP90.

We have added additional information about MMP8 in the introduction and discussion. We chose to analyse MMP8 as various studies have identified it as a protein with sex-related activation modulated by estrogens. Description of ERK densitometry was added to the legends and methods. Figure 5 was updated. The blots of pSMAD2 and pSMAD3 proved challenging; the submitted ones are our best.

Figure 7: The data provided in the manuscript do not accurately support the graphic. Male mice also have estrogen receptors that were omitted in the figure.

Figure 7 has been updated.

All figure legends should include type of ANOVA performed (two-way vs. one-way).

The legends have been updated.

Reviewer 3 Report

Review of manuscript ijms-1166675 entitled: Sex-related differences in murine models of chemically induced pulmonary fibrosis  by Pavel Solopov , Ruben Manuel Luciano Colunga Biancatelli , Christiana Dimitropoulou , John D. Catravas

The manuscript is the interesting and comprehensive. The experiments are well done. The study is devoted to investigation of gender effects on the lung fibrosis development. The experiments were performed on two lungs fibrosis models: Intratracheal application of HCl or  Nitrogen Mustard (NM) administration. The fibrosis was induced in male and female mice. Male mice exerted higher mortality and increased body weight loss. Moreover, in male subject the increased count of white blood cells and protein level in bronchoalveolar lavage fluid as well as more advanced fibrosis of the lungs and higher Ashcroft score were observed. Higher activation of TGF-beta (?) and signaling pathways (ERK, SMAD) was found in male subjects. Lung mechanics measurements were also performed. The Authors conclude that gender dependent effects of lung fibrosis are related to differences of TGF beta (?) and downstream signaling activation.

The manuscript could be interesting for the readers of International Journal of Molecular Sciences.

There are a few concerns with the study:

Minor concerns

  1. What isoform of TGF-ß (1-3) was investigated. The divergent effects (profibrotic and antifibrotic) of different isoforms of this growth factor were discussed in literature. What method was used for this cytokine assessment?
  2. The collagen type I was measured in the lungs. Why this type of collagen was selected?
  3. Description of methods of intratracheal catether insertion is not sufficiently clear.
  4. What is a justification for IL-1a and b evaluation. There are more cytokines involved in fibrosis regulation.

Author Response

Minor concerns

What isoform of TGF-ß (1-3) was investigated. The divergent effects (profibrotic and antifibrotic) of different isoforms of this growth factor were discussed in literature. What method was used for this cytokine assessment?

TGF-B1 was analysed in BALF by ELISA as described in the methods. We additionally included data of TGF-B isoform 1 analyzed by real-time qPCR in lung tissue homogenates.

The collagen type I was measured in the lungs. Why this type of collagen was selected?

Collagen type I was analyzed by western blotting. We choose collagen I because it is the most abundant collagen expressed in lung fibrosis.

Description of methods of intratracheal catheter insertion is not sufficiently clear.

These methods have been updated.

What is a justification for IL-1a and b evaluation. There are more cytokines involved in fibrosis regulation. We have previously analyzed a wide range of cytokines during HCl and NM instillation (https://www.tandfonline.com/doi/full/10.1080/08958378.2019.1624895; https://pubmed.ncbi.nlm.nih.gov/32362214/). Here, we have performed the analysis of cytokines that showed a significant increase after HCl and NM instillation.

Round 2

Reviewer 1 Report

Justification for studying the whole tissue lysates alone and not the cells for the mechanism is not acceptable.

Author Response

The reviewer does not consider the studies that we performed on lung homogenates adequate and suggests that we should perform these experiments in fibroblasts and/or macrophages isolated from lungs of male and female mice treated with HCl and nitrogen mustard, plus appropriate controls. This is a formidable undertaking that will necessitate treatments of new experimental groups and will take 6-8 weeks. Furthermore, we will have to develop the technique of isolating and characterizing macrophages and/or fibroblasts from mouse lung, since the methodology is not currently available in our lab. Given all the work that is already contained in our manuscript, we do not believe that performing these additional studies is justified. Furthermore, we and others have published in this Journal and elsewhere comparable studies where similar experiments were performed in lung homogenates rather than isolated fibroblasts or macrophages (https://pubs.rsc.org/en/content/articlehtml/2019/ra/c8ra08659a 

https://www.mdpi.com/1422-0067/21/13/4740).

Reviewer 2 Report

  1. Please check references - some do not correspond to the paper mentioned
  2. FlexiVent methods need to be clarified further. Was a deep inflation performed after the PV loops to reset the lung? The authors said for the snapshot-150 and quick prime they took the mean of 12 measurements per dose - of what?
  3. FlexiVent results are mentioned in the discussion, but not discussed/elaborated; please add significance
  4. The p-SMAD2 and p-SMAD3 westerns are difficult to see
  5. Figure 3A: the WBC label and justification by authors is inadequate. Unless differentials were done, there could be cells other than RBC and WBC
  6. In figure 4, two lanes for p-ERK and ERK, have three bands. Were the bands averaged or were they combined?
  7. A reference for other isoforms of TGFB might be helpful. The authors could also add data looking at the isoforms 

Author Response

  1. Please check references - some do not correspond to the paper mentioned

The references have been checked and corrected.

  1. 2. FlexiVent methods need to be clarified further. Was a deep inflation performed after the PV loops to reset the lung? The authors said for the snapshot-150 and quick prime they took the mean of 12 measurements per dose - of what?
  2. a) Yes; a deep inflation was performed both before and after each PV loop.
  3. b) Sorry; the “per dose” statement was left from studies of airway reactivity to methacholine, which are not included in this study and has now been removed. Both points are now clarified in Methods.
  4. FlexiVent results are mentioned in the discussion, but not discussed/elaborated; please add significance

Additional discussion about FlexiVent data has been added to the manuscript

  1. The p-SMAD2 and p-SMAD3 westerns are difficult to see.

We repeated the p-SMAD2 in NM samples and updated the figure. Unfortunately, we have run out of all HCl samples and cannot reproduce the blots.

  1. 5. Figure 3A: the WBC label and justification by authors is inadequate. Unless differentials were done, there could be cells other than RBC and WBC

We have changed WBC to total cell count in BALF.

  1. 6. In figure 4, two lanes for p-ERK and ERK, have three bands. Were the bands averaged or were they combined?

We quantified the combined top two bands only.

  1. 7. A reference for other isoforms of TGFB might be helpful. The authors could also add data looking at the isoforms 

We have added references about TGFb isoforms to the discussion. As mentioned above, we have run out of samples that would allow us further analyses, including TGFb isoforms.

Round 3

Reviewer 2 Report

none